# Which LLMs are Difficult to Detect? A Detailed Analysis of Potential Factors Contributing to Difficulties in LLM Text Detection

**Shantanu Thorat**
University of Cambridge
st980@cam.ac.uk

**Tianbao Yang**
Texas A&M University
tianbao-yang@tamu.edu

## Abstract

As LLMs increase in accessibility, LLM-generated texts have proliferated across several fields, such as scientific, academic, and creative writing. However, LLMs are not created equally; they may have different architectures and training datasets. Thus, some LLMs may be more challenging to detect than others. Using two datasets spanning four total writing domains, we train AI-generated (AIG) text classifiers using the LibAUC library - a deep learning library for training classifiers with imbalanced datasets. Our results in the Deepfake Text dataset show that AIG-text detection varies across domains, with scientific writing being relatively challenging. In the Rewritten Ivy Panda (RIP) dataset focusing on student essays, we find that the OpenAI family of LLMs was substantially difficult for our classifiers to distinguish from human texts. Additionally, we explore possible factors that could explain the difficulties in detecting OpenAI-generated texts.

## 1   Introduction

Large language models (LLMs) can generate human-like texts instantly. The increased accessibility of LLM models through services such as OpenAI's ChatGPT and AWS Bedrock has led to the proliferation of AI-generated texts (AIG-texts) on the web, particularly in scientific writing and online forum writing. Academic concerns about students' use of LLMs to complete assignments are also present. There are several LLMs — ranging from Meta's Llama to BigScience Bloom models — each with different architectures and training datasets that can cause differences in text generation.

Previous work on AI text detection has focused on evading detection (3) and evaluating detection work on out-of-distribution texts (4). However, to our knowledge, little work has been done on analyzing *which* LLMs are challenging to detect. This paper explores how LLM detection performance can vary across domains and even within families of LLMs.

We evaluate if the most challenging LLMs vary between writing domains using a subset of the Deepfake Text dataset (4). We also create a new essay dataset through the AWS Bedrock and OpenAI APIs by modifying a sample of student essays (9). We train classifiers on each dataset by optimizing the Area Under the Curve (AUC) metric to account for the imbalance of the training dataset. The AUC metric also considers the false positive rate, which is crucial as in some fields, such as education, false accusations of AI use can be costly. To this end, we use the LibAUC library for deep AUC maximization (10). We publish our source code, trained models, and datasets used at the following link.

38th Conference on Neural Information Processing Systems (NeurIPS 2024) - Safe Generative AI Workshop.

## 2 Datasets and Classifier

### 2.1 Deepfake Text Dataset

The Deepfake Text Detection dataset has over 400,000 texts spanning ten writing domains from 27 LLMs. This dataset is a binary classification dataset — texts are classified as "human" or "AI-generated." Table 1 lists all LLM families and their models.

Table 1: LLMs used to create AIG-texts in the Deepfake Text dataset.

| Family | LLMs |
| --- | --- |
| Llama | 7B, 13B, 30B, 65B |
| GLM | GLM130B |
| BigScience | T0-3B,T0-11B, BLOOM-7B1 |
| FLAN | small, base, large, xl, xxl |
| OpenAI | text-davinci-002,text-davinci-003, gpt-turbo-3.5 |
| OPT | 125M, 350M, 1.3B, 2.7B, 6.7B, 13B, 30B, iml-1.3B, iml-30B |
| EleutherAI | GPT-J-6B, GPT-NeoX-20B |

The three subsets we selected span the following writing domains: opinion statements (from Reddit's Change My View, CMV), scientific writing (Scigen), and story generation (from Reddit's Writing Prompts, WP). This dataset already provides a train-validation-test split. The distribution of training and test data within each writing domain can be seen in Table 2.

Table 2: Train-test split for the three domains.

| Domain | Train (Human/AIG) | Test (Human/AIG) |
| --- | --- | --- |
| CMV | 4,223/20,388 | 2,403/2,514 |
| Scigen | 4,436/18,691 | 2,538/2,251 |
| WP | 6,536/24,803 | 3,099/3,137 |

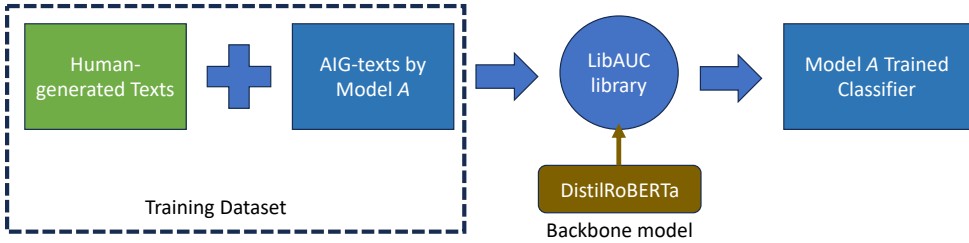

Figure 1: A framework of training a classifier to detect AIG-texts. We vary the model $A$ from different model families as shown in Table 1.

### 2.2 Rewritten Ivy Panda (RIP) Dataset

A drawback of the Deepfake dataset is that the number of texts from each LLM varies. We attempt to remove the effect of training dataset size by producing a balanced dataset. Additionally, we want to investigate a potential adversarial attack where users will intentionally prompt LLMs to generate human-like texts that can fool detectors. The RIP Dataset is a collection of rewritten AIG student writing essays from the Ivy Panda essay dataset (9). Essays were generated via a rewritten attack (7) by prompting eight LLMs — **Anthropic Claude Haiku** and **Sonnet**, **Meta Llama 2 13B** and **70B**, **OpenAI GPT-3.5** and **GPT-4o**, as well as **Mistral 7B** and **8x7B**. Here is an example prompt used to generate essays from the GPT-4o model:

> Please rewrite the essay and imitate its word using habits:

{human-written essay goes here}
Try to be different from the original essay.
Revised Essay:

The training set covers 9,000 human essays and 1,000 essays from each LLM for a total of 17,000 texts. The test set has 1,000 human essays and 125 essays from each LLM.

We use Amazon Bedrock and OpenAI's APIs to generate texts using different LLMs. The Bedrock and OpenAI APIs allow two parameters for prompting — temperature and top $p$ (1) (6). Temperature corresponds to how deterministic the LLM's output should be. Higher temperature values mean the LLM is more likely to generate a "wilder" output. The top $p$ value dictates the proportion of candidates to be selected during token generation. For example, a top $p$ value of 0.7 indicates that the model will consider the top 70% of tokens during generation. Regardless of LLMs, we select a temperature and a top $p$ value from a random uniform distribution from $[0.4, 1]$.

## 2.3 Using LibAUC to Train Classifiers

Since we need to process texts, we use a transformer-based model to build the classifier. Specifically, we use DistilRoBERTa (in the Huggingface transformers library, referred to as `DistilRoBERTaForSequenceClassification`). DistilRoBERTa is a distilled version of RoBERTa and is trained identically to DistilBERT (the down-sized version of BERT). It has 82M parameters, which suits our tasks as we need to train many classifiers.

For the Deepfake Text dataset, we train individual classifiers on the three domains (CMV, Scigen, WP). Within each domain, we train classifiers with human-written texts and AIG texts from one LLM, as illustrated in Figure 1. We also train one classifier on the entire training corpus within each domain (refer to these classifiers as *super-classifiers*). Thus, we have 28 classifiers for each domain, and hence a total of 84 classifiers. On the RIP Dataset, we train eight classifiers, one for each LLM's set of generated texts.

As a result, we train a total of 92 classifiers. To train each classifier, we augment DistilRoBERTa with a classifier head that is randomly initialized and fine-tuned all model parameters. Since the number of human-created texts and AIG-texts are unequal, we account for the data imbalance by directly optimizing AUC. Each classifier is trained using the LibAUC's `CompositionalAUCLoss` function (learning rate = 0.02) and `PDSCA` optimizer. The LibAUC CompositionalAUCLoss is built on top of the AUCMLoss function as well as the standard cross entropy loss function as defined below:

$$L_{\text{AUC}} \left( \mathbf{w} - \alpha \nabla L_{\text{CE}}(\mathbf{w}) \right) \tag{1}$$

Since the loss function needs at least one positive (AI-generated) and one negative (human-written), we used a `DualSampler` while training with a sampling rate of 0.5 for each mini-batch (thus, each mini-batch has a balanced amount of AIG-texts and human texts). The batch size was 32 across all training experiments. Each classifier was trained for one epoch. The training was done on the Kaggle platform, using GPU T4 $\times$ 2. The PyTorch Dataset wrapper used in training was adapted from the experiments used to detect fake reviews by Salminen et al (8).

## 3 Results

### 3.1 Deepfake Results

#### 3.1.1 Super-Classifier Results

We stratify the test corpus by LLM and then create 27 test sets within each domain; each test set contains all the human-written texts and texts from one LLM.

We first show the performance of the super-classifier that is trained by using all human-created texts and AIG texts within each domain and then evaluate the classifiers on the 27 test sets, each composed of all human-written texts plus one LLM's texts. The averaged results are shown in Table 3. We can see that LLM-text detection in scientific writing is slightly more difficult than in opinion statements and story generation. However, the variance is higher across all LLMs within scientific writing. CMV has a much lower variance than the other two writing domains, suggesting that detection performance in opinion statements is more consistent across different LLMs.

Table 3: Summary of mean AUC scores of super-classifiers on 27 subsets with standard deviation.

| Domain | Mean AUC Score (Mean $\pm$ SD) |
|---|---|
| CMV | $0.988 \pm 0.006$ |
| Scigen | $0.975 \pm 0.025$ |
| WP | $0.985 \pm 0.011$ |

### 3.1.2 Individual Classifier Results

For the other 81 classifiers, we aggregate the performance according to the LLM family from two dimensions, with one dimension corresponding to LLMs used for generating training data and another corresponding to LLMs used for generating the testing data. An illustration of aggregating the performance of classifiers using the AIG texts of OpenAI LLMs is shown in Figure 2. Results are shown in tables 4, 5 and 6 for the three domains. **Bolded values** indicate the highest AUC obtained for a test set (i.e., column); underlined values indicate the highest AUC obtained by a group of classifiers where the test set LLM family was not the same as the training set's family.

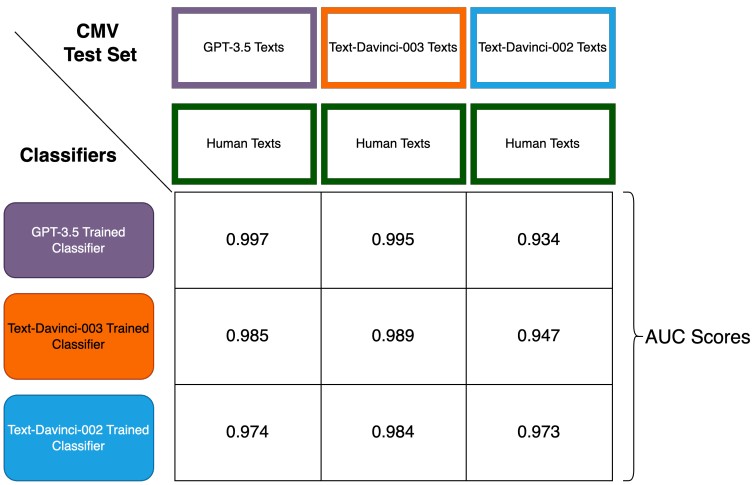

Figure 2: CMV testing framework for evaluating OpenAI-trained classifiers' performance on OpenAI texts. We have three OpenAI classifiers with three different possible test sets leading to nine AUC scores. The mean AUC score for OpenAI classifiers on OpenAI texts was 0.976. This testing procedure was repeated across all combinations of LLM families.

Table 4: Mean AUC scores computed as seen in Figure 2 within the CMV test subset.

| LLM Family Training Set | LLM Family Test Set | | | | | | |
|---|---|---|---|---|---|---|---|
| | BigScience | EleutherAI | FLAN | GLM | Llama | OpenAI | OPT |
| BigScience | 0.830 | 0.643 | 0.924 | 0.394 | 0.375 | 0.791 | 0.793 |
| EleutherAI | 0.805 | 0.900 | 0.763 | 0.539 | 0.540 | 0.677 | 0.862 |
| FLAN | 0.824 | 0.618 | **0.961** | 0.516 | 0.492 | 0.857 | 0.786 |
| GLM | 0.800 | 0.520 | 0.895 | **0.995** | 0.979 | 0.918 | 0.701 |
| Llama | 0.659 | 0.581 | 0.661 | 0.969 | 0.952 | 0.718 | 0.616 |
| OpenAI | 0.873 | 0.632 | 0.952 | 0.649 | 0.599 | **0.976** | 0.826 |
| OPT | **0.934** | **0.905** | 0.938 | 0.529 | 0.527 | 0.850 | **0.933** |

Table 5: Mean AUC scores computed as seen in Figure 2 within the Scigen test subset.

| LLM Family Training Set | LLM Family Test Set | | | | | | |
|---|---|---|---|---|---|---|---|
| | BigScience | EleutherAI | FLAN | GLM | Llama | OpenAI | OPT |
| BigScience | 0.667 | 0.594 | 0.745 | 0.605 | 0.594 | 0.574 | 0.567 |
| EleutherAI | 0.901 | **0.963** | 0.910 | 0.777 | 0.583 | 0.557 | 0.864 |
| Flan | 0.770 | 0.591 | **0.980** | 0.707 | 0.635 | 0.677 | 0.552 |
| GLM | **0.965** | 0.862 | 0.979 | **0.957** | 0.872 | 0.771 | 0.764 |
| Llama | 0.885 | 0.713 | 0.922 | 0.939 | **0.885** | 0.805 | 0.727 |
| OpenAI | 0.778 | 0.812 | 0.842 | 0.760 | 0.688 | **0.852** | 0.828 |
| OPT | 0.850 | 0.954 | 0.819 | 0.795 | 0.633 | 0.722 | **0.928** |

Table 6: Mean AUC scores computed as seen in Figure 2 within the WP test subset.

| LLM Family Training Set | LLM Family Test Set | | | | | | |
|---|---|---|---|---|---|---|---|
| | BigScience | EleutherAI | FLAN | GLM | Llama | OpenAI | OPT |
| BigScience | **0.947** | 0.792 | 0.991 | 0.605 | 0.586 | 0.836 | 0.811 |
| EleutherAI | 0.947 | 0.927 | 0.948 | 0.584 | 0.522 | 0.770 | *0.909* |
| FLAN | 0.872 | 0.618 | 0.986 | 0.626 | 0.574 | 0.743 | 0.632 |
| GLM | 0.893 | 0.497 | **0.997** | **1.000** | **0.995** | 0.959 | 0.721 |
| Llama | 0.862 | 0.532 | 0.930 | 0.994 | 0.988 | 0.943 | 0.721 |
| OpenAI | 0.908 | 0.606 | 0.986 | 0.873 | 0.835 | **0.985** | 0.777 |
| OPT | 0.945 | **0.959** | 0.928 | 0.600 | 0.592 | 0.790 | **0.949** |

We have the following observations from the results: (1) the classifier trained on texts generated by one model from an LLM family usually performs well on texts generated by the same family of LLMs; (2) across all writing domains, Llama and GLM texts are relatively hard to detect unless using the classifiers that are trained on texts from those two families; (3) OpenAI LLMs also pose a challenge, especially in the Scigen subset.

## 3.2 RIP Results

We aggregate the performance in a manner similar to that of the Deepfake Text dataset. The results are shown in Figure 3. We can see that regardless of which LLM texts were used in training, all classifiers, except for OpenAI-trained classifiers, struggle significantly with the AIG texts from OpenAI's model family. The Claude Haiku-trained classifier was the only non-OpenAI-trained classifier to achieve an AUC score of at least 0.900 on the OpenAI test sets. In contrast, the remaining six LLM test sets have at least one classifier, not trained on an LLM from the same test set's family, achieving an AUC score of at least 0.990.

## 3.3 More Analysis on the RIP Dataset

In this subsection, we perform some analysis to explore possible factors contributing to the difficulty in detecting LLM-generated texts.

*Entropy.* We compute Shannon entropy of a given text $X$ (where $x_i$ is the $i$-th token of $X$) as follows (2):

$$H(X) = -\sum_{i=1}^{n} p(x_i) \ln p(x_i) \qquad (2)$$

$p(x_i)$ is the probability of token $x_i$ from a large English corpus (5). Higher entropy values indicate a higher complexity of text. In Figure 4, we calculate a kernel density estimate (KDE) on the entropy distributions for texts from the four LLM families (we include "human" as a distinct family for comparison) in the test set. Human texts are substantially more likely to have higher entropies. Claude, Llama2, and Mistral have similar distributions of entropy. In contrast, the OpenAI entropy distribution is shifted rightwards closer to the human texts. This shift indicates that OpenAI texts are slightly more complex than other LLM families, making them more difficult to detect.

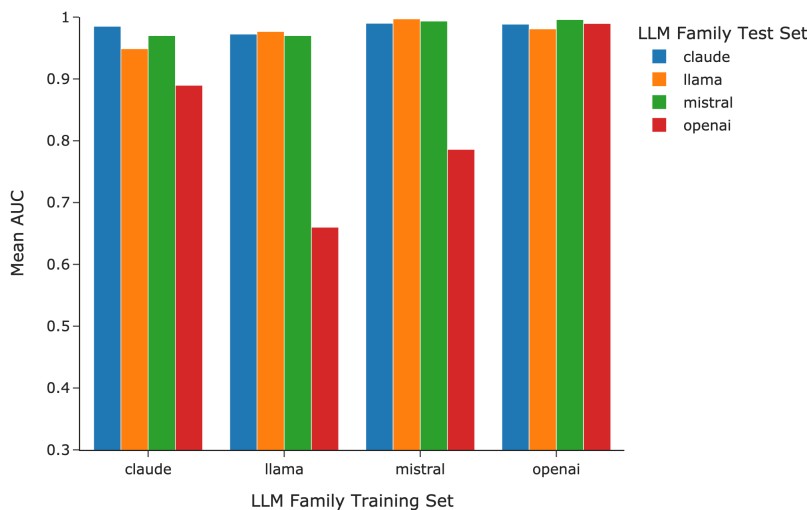

Figure 3: Mean AUC by LLM family on the RIP Bedrock dataset. Mean AUC is computed identically to the Deepfake dataset in Figure 2.

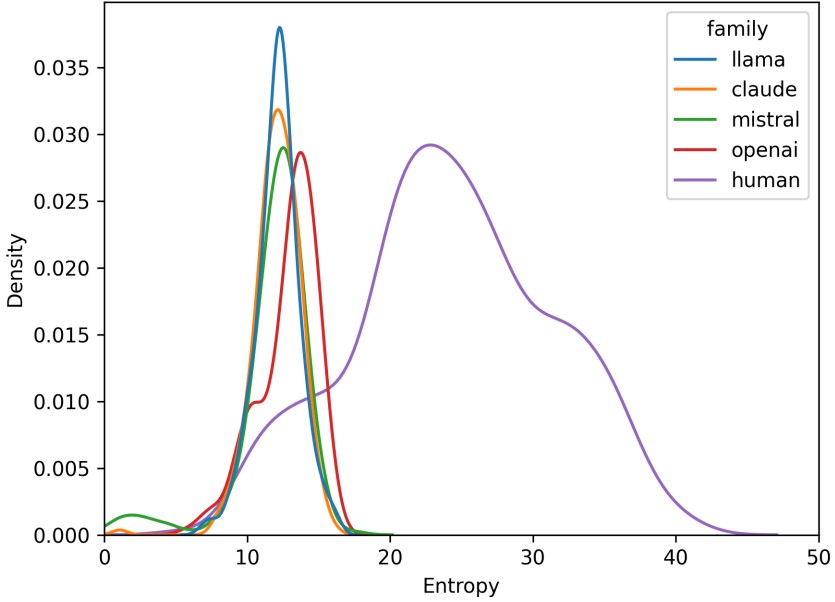

Figure 4: Kernel density estimates of the entropy distributions for the four LLM families — Claude, Llama2, Mistral, and OpenAI — from the test set. The KDE for entropy in human-authored essays is included as a baseline.

*Out of vocabulary ratio.* Out of vocabulary (OOV) ratio is computed as the number of tokens not in the spaCy vocabulary out of all tokens for a given text (2).

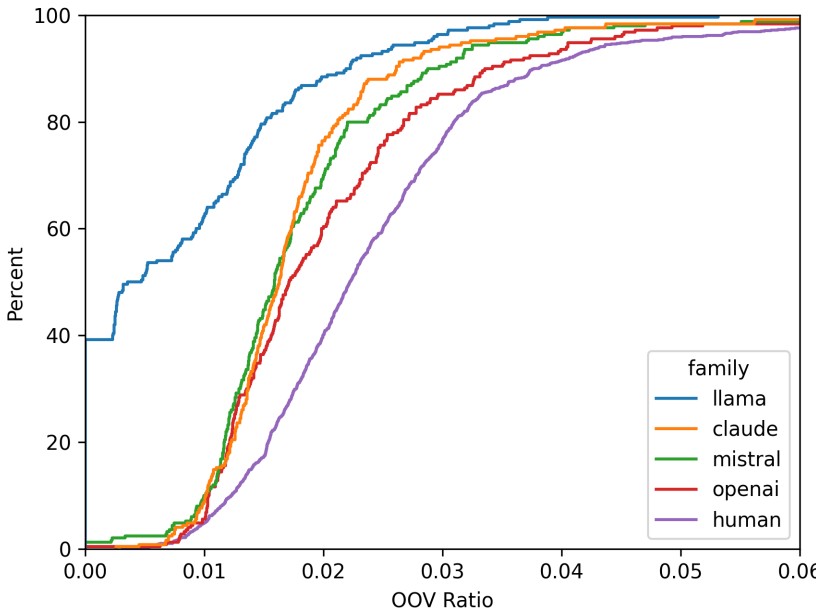

Figure 5: The empirical cumulative distribution function plots for the four LLM families — Claude, Llama2, Mistral, and OpenAI — from the test set. The ECDF for human-authored essays is included as a baseline.

In figure 5, we observe that the OpenAI texts are more similar to human texts regarding the OOV ratio distribution. A possible reason for the high entropy observed in the human texts (as seen in figure 4) is due to human essays using words not found in the spaCy vocabulary. OpenAI LLMs might be "borrowing" terms from human essays more frequently when asked to rewrite them when compared to other LLMs.

## 4  Conclusion

In this paper, we have conducted a fine-grained analysis of detecting AI-generated texts by different LLMs using classifiers trained from texts generated by different LLMs. In the Deepfake dataset, LLM detection difficulty varies significantly across writing domains. Story generation (WP) has a high floor for AIG-text detection — the "worst" performance was on the BigScience family test set(mean AUC score of 0.947). In contrast, scientific writing is more challenging, with the lowest mean AUC score (0.852) on the OpenAI test set. Opinion writing (CMV) detection is less challenging than detection in scientific writing, with the lowest mean AUC score of 0.905 (on the EleutherAI test set). Regarding student essay detection, we found that the OpenAI LLMs were the most difficult to distinguish from humans, *except* by classifiers trained on OpenAI texts. Further investigation demonstrated that OpenAI texts are similar to human texts based on entropy and OOV ratio.

These results suggest that the quality of AIG texts varies between writing domains. Training a classifier with one LLM (or even one family) also leads to some generalizability in detecting texts from out-of-distribution families. However, we limit our analysis to writing domains that typically involve longer texts. For example, Amazon reviews and social media posts are often shorter. Future work might explore how LLM detection varies across these domains, which are highly susceptible to LLM-generation abuse.

Future AI-text detector developers should take care in curating a diverse corpus of texts spanning several domains and LLMs and take note of which LLMs are better at mimicking human writing.

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

## A  Appendix / supplemental material

The following tables include individual classifiers' AUC scores on specific LLM model test sets for the RIP dataset. Complete results on the Deepfake dataset can be found at our code repository.

**Bolded values** indicate the highest AUC obtained for a test set (i.e., column); underlined values indicate the highest AUC score obtained by a classifier trained on a different LLM family from the test set.

Table 7: AUC scores of individual classifiers (Claude and OpenAI test sets) on the RIP dataset.

| LLM Family Training Set | LLM Family Test Set | | | |
|---|---|---|---|---|
| | Claude Haiku | Claude Haiku | GPT-3.5 | GPT-4o |
| Claude Haiku | **1.000** | **0.999** | 0.956 | 0.901 |
| Claude Sonnet | 0.962 | 0.979 | 0.892 | 0.809 |
| GPT-3.5 | 0.998 | 0.958 | 0.989 | 0.975 |
| GPT-4o | **1.000** | 0.998 | **0.994** | **1.000** |
| Llama2-13B | 0.938 | 0.955 | 0.709 | 0.556 |
| Llama2-70B | 0.999 | 0.998 | 0.785 | 0.590 |
| Mistral 7B | 0.981 | 0.983 | 0.868 | 0.809 |
| Mistral 8x7B | 0.999 | 0.997 | 0.837 | 0.629 |

Table 8: AUC scores of individual classifiers (Llama2 and Mistral test sets) on the RIP dataset.

| | LLM Family Test Set | | | |
|---|---|---|---|---|
| LLM Family Training Set | Llama2-13B | Llama2-70B | Mistral 7B | Mistral 8x7B |
| Claude Haiku | 0.964 | 0.958 | 0.999 | 0.990 |
| Claude Sonnet | 0.996 | 0.877 | 0.954 | 0.936 |
| GPT-3.5 | 0.970 | 0.989 | 0.995 | 0.991 |
| GPT-4o | 0.987 | 0.977 | **1.000** | **0.999** |
| Llama2-13B | 0.964 | 0.948 | 0.954 | 0.928 |
| Llama2-70B | 0.999 | 0.996 | **1.000** | **0.999** |
| Mistral 7B | **1.000** | 0.993 | 0.991 | 0.988 |
| Mistral 8x7B | **1.000** | **0.995** | 0.998 | 0.997 |

