# OpenReview forum: "Which LLMs are Difficult to Detect? A Detailed Analysis of Potential Factors Contributing to Difficulties in LLM Text Detection"
_NeurIPS.cc/2024/Workshop/SafeGenAi — SafeGenAi Poster_

### Official Review · Reviewer_Y94f · 2024-10-09
**The works present an exhaustive set of experiments for comparing performance of AI generated text detection across different datasets on state-of-the-art LLMs .**

**Rating:** 9
**Confidence:** 4

**Review:**

The work demonstrates how LLM text detection varies between text domains and across LLMs families. An extensive analysis is presented  to detect AI based text using classifiers  trained on text from different classifiers. This is a first work evaluating and comparing the AI detection across various LLMs, and demonstrates how some LLMs generated text is harder than the others, specially when the theme of the dataset changes.
For the classification task,  DistilRoBERTa, a transformer based model was used. A separate classifier is build for each data domain (story generation, science writing, and reddit's change my view) for each LLM (27).

The paper presents a strong and exhaustive experimental analysis on comparing various LLMs. This analysis which is first of it's kind. The authors not only demonstrate the family of LLMs which is hardest to distinguish from human text but also provide insights into the underlying structure of this observation. Hence,  I recommend strong accept for this work.

**Strengths**
1. The work demonstrates experiments on an extensive set of state-of-the-art LLMs as well as multiple datasets.
2. Authors provide a very insightful observation on OpenAI texts being similar to human authored texts in terms of entropy and out of vocabulary ratio which makes Open AI LLMs text detection much harder. This insight is very helpful in designing defense strategies that are needed to protect against AI generated texts.

**Areas of improvements**
1. Paper would benefit from analysis of the effect of temperature in a controlled manner which currently is randomly sampled.
2. The paper currently does not include datasets comprising shorter texts. To ensure the completeness and robustness of this work, it would be beneficial to incorporate an additional dataset featuring shorter texts.

---

### Official Review · Reviewer_AXDY · 2024-10-09
**Good paper but need more analysis according to your experiment**

**Rating:** 6
**Confidence:** 5

**Review:**

This paper investigates the difficulties in detecting AI-generated texts produced by different large language models (LLMs). The authors train AI-generated text classifiers using two datasets and analyze the performance of the classifiers across different writing domains and LLM families. They find that LLM-text detection varies across domains, with scientific writing being more challenging. Additionally, they observe that OpenAI LLMs are particularly difficult to distinguish from human texts in the context of student essays. The paper also explores possible factors contributing to the difficulties in detecting OpenAI-generated texts.

1. Could you provide more insights into the practical implications of the difficulties in detecting AI-generated texts? How can the findings of this study be applied to improve detection methods?
2. Can you further elaborate on the factors that contribute to the difficulties in detecting OpenAI-generated texts? Specifically, could you provide more detailed analysis and discussion on the similarities between OpenAI texts and human texts based on entropy and OOV ratio?
3. Have you considered exploring how LLM detection varies across domains that involve shorter texts, such as social media posts or online reviews? This could provide further insights into the challenges of LLM-text detection in different contexts.
4. Could you clarify the novelty of your research? While the investigation of LLM-text detection difficulties is important, it would be helpful to highlight the unique contributions of your study to the existing literature.
5. Could you provide more details about the hyperparameters used for training the classifiers, such as learning rate and optimizer? Additionally, what was the reasoning behind choosing these specific hyperparameters?
6. The paper does not discuss potential ethical implications or risks associated with the detection of AI-generated texts. Could you briefly address these concerns and discuss any safeguards that could be put in place to mitigate these risks?